# Functional Outcomes Following Hip Replacement in Community-Dwelling Older Adults

**DOI:** 10.3390/jcm11175117

**Published:** 2022-08-30

**Authors:** Yuanyuan Wang, Alice Owen, Angus Franks, Ilana Ackerman, Sharyn M. Fitzgerald, Susan Liew, Robyn L. Woods, Anita E. Wluka, John J. McNeil, Flavia M. Cicuttini

**Affiliations:** 1School of Public Health and Preventive Medicine, Monash University, Melbourne 3004, Australia; 2Alfred Hospital, Melbourne 3004, Australia

**Keywords:** hip replacement, older adults, physical function

## Abstract

Uncertainty remains regarding the benefit of hip replacement in older adults in the context of age-related decline in physical function. This study aimed to examine the effect of hip replacement on functional outcomes and identify factors associated with clinically important improvement in physical function postoperatively in community-dwelling older adults. This cohort study was performed within the ASPREE trial, with 698 participants receiving hip replacement and 677 age- and sex-matched controls without knee or hip replacement during the trial drawn from 16,703 Australian participants aged ≥70 years. Health status (physical and mental component summary [PCS and MCS]) was assessed annually using the SF-12. Participants receiving hip replacement had significantly lower pre- and post-replacement PCS scores compared with controls (*p* < 0.0001). There was significant improvement in PCS score following hip replacement (mean change 4.9, 95%CI 4.0–5.7) but no change in controls (0.01, 95%CI −0.7–0.7). Following hip replacement, 46.7% of participants experienced clinically important improvement in PCS score, while 15.5% experienced worsened PCS score. Participants experiencing improved postoperative PCS score had significantly lower PCS and higher MCS scores preoperatively. The degree of preoperative physical function impairment was a significant indicator of older people most likely to benefit from hip replacement surgery.

## 1. Introduction

Hip osteoarthritis (OA) causes significant pain and disability. There is no approved disease-modifying treatment that slows disease progression, with end-stage hip OA being managed with hip replacement. The median age for total hip replacement in Australia is 69 years, reflecting that a significant proportion of procedures are undertaken in older adults [1]. Hip OA differs from knee OA, the other common reason for total joint replacement, as there tends to be faster progression of hip OA [2] and thus a shorter time from the onset of symptoms to the need for hip replacement for OA [2,3], with an average of 8 years for the hip and 16 years for the knee [3]. Although both total knee and total hip replacements have been found to confer substantial improvement in functional outcomes, greater improvement in physical function has been reported in patients receiving total hip replacement [4,5,6,7,8,9], resulting in lower dissatisfaction rates in patients receiving total hip replacement compared with those receiving total knee replacement (5–15% [7,10,11,12,13] vs. 8–25% [14]).

Total hip replacement is one of the most successful surgical procedures, providing excellent pain relief and function restoration for patients with severe hip OA [15]. The incidence of total hip replacement is projected to increase significantly over the next decades [16,17]. However, this is a major surgical procedure associated with increased risk of postoperative morbidity and mortality, particularly in older populations [18,19]. Despite almost half of all total hip replacements for OA being performed in older adults (those aged 70 years and over), the lack of data on functional outcomes following hip replacement in this older population means that outcomes are extrapolated from data from younger populations or studies involving both younger and older patients. However, the benefits of hip replacement in community-based older adults need to be determined in the context of age-related decline in physical function and other comorbidities [20,21]. This information is currently lacking but is important to provide the evidence needed to assist decision making about the risk–benefit comparison of this major orthopedic surgery in older adults. Although it is costly and associated with complications, it has the potential to significantly improve physical function and quality of life, and identifying those most likely to benefit is important.

Large-scale cohort studies provide an opportunity to assess functional outcomes following joint replacement compared to age- and sex-matched controls without joint replacement. This enables the effect of joint replacement in older adults to be examined in the context of age-related decline in physical function which is increased after the age of 70 years [20,21]. Thus, the aims of the present study were to (1) examine the effect of hip replacement on functional outcomes in a cohort of community-dwelling adults aged ≥70 years, compared to age- and sex-matched controls without joint replacement; and (2) identify factors associated with minimal clinically important improvement in self-reported physical function after hip replacement. 

## 2. Materials and Methods

### 2.1. Study Population and Setting

This cohort study was conducted within the ASPirin in Reducing Events in the Elderly (ASPREE) trial with study design, participant characteristics and primary results published previously [22,23,24]. In brief, ASPREE was a randomized, placebo-controlled trial determining whether 100 mg/day aspirin extended disability-free survival over 5 years in 19,114 community-dwelling adults from Australia (aged ≥70 years) and the USA (aged ≥65 years). Individuals with diagnosed cardiovascular disease, dementia or physical disability (major difficulty with performing independently any one of six basic activities of daily living) were excluded, resulting in a relatively healthy, independently-living cohort at trial enrolment. The median follow-up period was 4.7 years. The study was approved by Ethics Committees in Australia and the USA and registered on Clinicaltrials.gov (NCT01038583). All participants provided written informed consent. The participants of the present study were drawn from the 16,703 trial participants from Australia. Australia has a universal healthcare system providing publicly funded access to joint replacement, with subsidized access also available through private health providers. 

### 2.2. Identification of Participants with Hip Replacement and Age- and Sex-Matched Controls

Clinical documentation relating to hospitalizations for all knee and hip surgical procedures during the ASPREE trial was reviewed, and participants with any hip replacement procedure (most with the indication recorded as OA; any hip replacements secondary to fractures were excluded) were identified, with the first recorded in-trial hip replacement being used to confer hip replacement status. Participants with hip replacement were matched 1:1 by age (±1 year) at time of hip replacement and sex with controls (age at study entry) who did not have a joint replacement during the ASPREE trial, as shown in Figure 1. Baseline was defined as the preoperative study visit for participants with hip replacement and the ASPREE baseline study visit for controls.

### 2.3. Outcome Measures

Health status was assessed annually using the Medical Outcomes Study Short Form 12 Health Survey Version 2 (SF-12v2) [25], a generic health profile instrument consisting of 12 items that measure 8 health domains to assess physical and mental health: physical functioning, role physical, bodily pain, general health, vitality, social functioning, role emotional, and mental health. Each domain has a score ranging from 0 (worst health) to 100 (best health). The scores of these 8 domains can be weighted and summarized into two composite scores, the physical component summary (PCS) score and the mental component summary (MCS) score, each having the range 0–100 with higher values indicating better health status [26].

Gait speed, a simple, objective indicator of physical function and mobility [27,28], was measured by research staff at the randomization visit and thereafter biennially, as the time in seconds to walk 3 m at the participant’s usual walking pace from a standing start, with a gait aid, if used. Time on the faster of two walks was taken as the final gait speed measure [29].

For cases, the pre-hip replacement measure was selected as the assessment undertaken at the study visit immediately prior to the hospitalization for hip replacement. The post-hip replacement measure was taken as the assessment undertaken at the first available study visit at least 6 months after the hospitalization for hip replacement. For controls, outcomes were examined at ASPREE study baseline, and then at 1 year for health status and 2 years for gait speed after randomization.

### 2.4. Covariates

Age, sex, weight, height, years of education, and morbidities or chronic conditions (diabetes mellitus, dyslipidemia, hypertension, chronic kidney disease, and cancer) were collected in the ASPREE study as previously described [22]. Body mass index (BMI) was calculated from body weight/height^2^. Self-reported history of joint replacement before study enrolment was obtained from a sub-study, the ASPREE Longitudinal Study of Older Persons [30].

### 2.5. Statistical Analysis

Characteristics of participants with hip replacement and age- and sex-matched controls were compared using independent samples t-tests or chi-square tests, when appropriate. For aim 1, i.e., comparisons of changes in functional outcomes between participants with hip replacement and controls, independent samples *t*-tests were used for unadjusted analyses, and multiple linear regression models were fitted with change in functional outcomes as the dependent variable and hip replacement vs. control as the independent variable, adjusted for baseline BMI, education, morbidities or chronic conditions, and time between outcome measures. For aim 2, participants with hip replacement were categorized into three groups based on the minimal clinically important difference in PCS score, which was 4.6 points after total joint replacement [31]: worsened (>4.6 reduction), stable (change ≥−4.6 and <4.6), and improved (≥4.6 increase) physical function. To compare participant characteristics among the physical function categories, analysis of variance was used for unadjusted analyses and general linear model was used to adjust for covariates.

Stratified analyses by age category and sex were performed. Sensitivity analyses were undertaken, excluding participants who self-reported a history of joint replacement at baseline. To reduce the risk of type 1 error, adjustment for multiple comparisons was undertaken and alpha was set at 0.005 for this study. All analyses were undertaken using STATA 17 (College Station, Texas USA).

## 3. Results

This study analyzed 698 participants with hip replacement and 677 age- and sex-matched controls (Figure 1). The characteristics of study participants at ASPREE baseline are presented in Table 1. The mean age was 77.9 years, with women over-represented in the cohort. Participants with hip replacement had significantly higher BMI, lower PCS score, and lower prevalence of chronic kidney disease than controls. The scores of physical function, role physical, and bodily pain were significantly lower in participants with hip replacement compared with controls. The MCS score, scores of other SF-12 domains, gait speed, and prevalence of diabetes, dyslipidemia, hypertension, and cancer did not differ significantly between the two groups. The median time between baseline and follow-up SF-12 measures was 410 days (range 282–1267 days) for participants with hip replacement and 362 days (range 232–906 days) for controls.

### 3.1. Hip Replacement and PCS

Participants with hip replacement had significantly lower PCS score prior to hip replacement compared with controls (39.8 vs. 48.2, *p* < 0.0001) (Table 2; Appendix A). In the analysis adjusted for BMI, education, morbidities or chronic conditions, and days between outcome measures, there was a significant increase in PCS score following hip replacement (mean change 4.9, 95% confidence interval (CI) 4.0, 5.7), while no significant change was observed for controls over that time (0.01, 95%CI −0.7, 0.7). There was a significant between-group difference (4.8, 95%CI 3.7, 6.0) (Table 2). Post-hip replacement PCS score for participants with hip replacement remained significantly lower than for controls (44.5 vs. 48.0, *p* < 0.0001) (Table 2; Appendix A). 

### 3.2. Hip Replacement and MCS 

The MCS score was slightly higher in participants with hip replacement prior to hip replacement compared with controls (56.9 vs. 55.8, *p* = 0.02) (Table 2; Appendix A). In the adjusted analysis, there was a modest decline in MCS score in participants with hip replacement following hip replacement (mean change −1.6, 95%CI −2.4, −0.8) which was not observed in controls (0.1, 95%CI −0.5, 0.8), with a significant between-group difference (−1.7, 95%CI −2.8, −0.7) (Table 2). The postoperative MCS score in participants with hip replacement did not differ significantly from the MCS score for controls at follow-up (55.6 vs. 56.0, *p* = 0.31) (Table 2; Appendix A). 

### 3.3. Hip Replacement and SF-12 Health Domains 

Prior to hip replacement, participants with hip replacement had significantly lower scores than controls in physical function (40.7 vs. 47.6, *p* < 0.0001), role physical (44.3 vs. 49.7, *p* < 0.0001), bodily pain (40.1 vs. 50.2, *p* < 0.0001), general health (49.4 vs. 52.2, *p* < 0.0001), vitality (52.1 vs. 54.8, *p* < 0.0001), social functioning (51.5 vs. 53.7, *p* < 0.0001), and role emotional (50.4 vs. 51.7, *p* = 0.006) (Table 2; Appendix A). Participants continued to have significantly lower scores in these domains after hip replacement compared with controls (all *p* ≤ 0.005) (Table 2; Appendix A). In the adjusted analyses, participants had significant improvement in physical function (mean change 3.8, 95%CI 2.9, 4.8), role physical (2.5, 95%CI 1.6, 3.4), and bodily pain (6.7, 95%CI 5.6, 7.7) following hip replacement, while controls experienced no significant change in any domains. Between-group differences were significant for physical function, role physical, and bodily pain (Table 2). 

### 3.4. Hip Replacement and Gait Speed 

Gait speed was significantly lower in participants with hip replacement prior to hip replacement compared with controls (0.96 vs. 1.05 m/s, *p* < 0.0001) (Table 2). In the adjusted analysis, gait speed significantly declined in controls (mean change −0.05 m/s, 95%CI −0.06, –0.03) but not in participants with hip replacement (−0.02 m/s, 95%CI −0.04, 0.01) over the follow-up period, with little between-group difference (0.03, 95%CI 0.0005, 0.06) (Table 2). Post-hip replacement gait speed in participants with hip replacement remained significantly lower than that of controls (0.95 vs. 1.01 m/s, *p* = 0.0001) (Table 2).

Similar results were observed for males and females. Female participants with hip replacement experienced a greater improvement in PCS, physical function, role physical, and bodily pain than male participants with hip replacement, when compared with the controls (Appendix A). Similar results were also seen in analyses stratified by age category. The results did not change in sensitivity analyses that excluded participants with self-reported joint replacement prior to the ASPREE trial (Appendix A).

### 3.5. Factors Associated with Clinically Important Change in PCS Score after Hip Replacement 

Baseline characteristics were compared among participants with hip replacement with worsened, stable, and improved PCS scores (Table 3). Following hip replacement, 208 (46.7%) participants experienced an improved PCS score, and 69 (15.5%) experienced a worsened PCS score. Participants who experienced an improved PCS score had significantly lower preoperative PCS, physical function, role physical, and bodily pain scores, and significantly higher preoperative MCS scores. There were no significant differences among the PCS status groups with respect to age, sex, education, BMI, morbidities or chronic conditions, scores of general health, vitality, social functioning, role emotional, and mental health, or gait speed. Similar results were observed in the adjusted analysis.

## 4. Discussion

In this large cohort of healthy community-dwelling older adults, although participants experienced a significant improvement in PCS scores after hip replacement, their postoperative physical function remained significantly lower than age- and sex-matched controls. The greatest improvements were observed for the bodily pain and physical function domains of the SF-12. A higher level of preoperative physical function impairment was the strongest predictor of clinically important improvement in PCS score after hip replacement. Those who were already functioning well were less likely to benefit from this major surgery, suggesting that the level of physical function could be used to inform decision making as to which older adults will have functional improvement following hip replacement surgery.

Functional improvement is an important patient-related outcome following hip replacement [4,5,6,7,8,9]. However, data are limited for the effect of hip replacement on functional outcomes in community-based older populations. In hospital and arthroplasty registry settings, there has been consistent evidence for the beneficial effect of total hip replacement on improving physical function and quality of life [4,5,6,7,8,9,32]. However, these data are subject to bias conferred by the selection of study populations, as these studies are from hospital settings with single or multiple healthcare provider sites. Furthermore, none of these studies provide outcome data on older adults (aged ≥70 years) who represent about half of those receiving a hip replacement. In contrast, our study examined community-dwelling older adults taking part in a large clinical trial who were selected on the basis of being relatively healthy and living independently at enrolment. Such participants represent the types of community-based older people who are candidates for hip replacement if medically indicated. The participants in our study received hip replacement surgery as part of routine healthcare and decision making, spanned both public and private health systems, and were not restricted to single clinical sites, therefore reflecting the real-world situation. This likely explains the higher preoperative PCS and MCS scores in our study compared to previous studies [4,5,6,7,8,9,32], suggesting that highly-selected study populations and clinical samples of patients seen in many studies do not represent average community-dwelling older individuals who undergo hip replacement surgery.

Age-related decline in physical function is commonly experienced in older people [20,21], so the impact of hip replacement on functional outcomes needs to be considered in this context. In this study, we examined functional outcomes in participants receiving hip replacement surgery, compared with age- and sex-matched controls with a similar likelihood of age-related functional decline but without joint replacement. This cannot be feasibly explored using large-scale arthroplasty registry data or in hospital-based clinical studies. In our study, participants with hip replacement had significantly lower PCS scores and gait speed prior to surgery compared with matched controls, reflecting the indications for and decision to undergo a hip replacement surgery. However, although a significant improvement in PCS scores was shown in participants following hip replacement, their postoperative PCS scores remained significantly lower than for controls with approximately half the preoperative differences persisting. In participants with hip replacement, the greatest improvements were seen in bodily pain and physical function, with clinically significant increases in mean scores of 17% and 8% after hip replacement, respectively. Although joint-specific pain was not assessed in our study, the significant improvement in bodily pain following hip replacement is likely to have contributed to the improvement in physical function. Previous meta-analysis showed moderate evidence of increased walking speed at 12 months after total hip replacement [33]. However, this was in younger people than in our study. In our study, the controls had a significant reduction in gait speed, but gait speed remained stable in participants with hip replacement, with a lower speed than the controls. Hip replacement seems to have halted the age-related decline in gait speed.

Consistent with results from previous studies [34,35], our study showed that improvement in physical function was less likely in those with better physical function and those with worse mental health function preoperatively. These data highlight the importance of careful selection of patients in order to identify those who are most likely to benefit from hip replacement surgery. Although symptomatic arthritis is very common in older adults, its impact on function can vary. As hip replacements are performed to improve physical function and independence in older adults, assessing the degree of physical functional impairment should be considered when identifying those who require hip replacement and are most likely to benefit. If older individuals have hip OA but are functioning well, a careful discussion is needed in order to clarify the therapeutic goals for a hip replacement. This is important when considering the risk–benefit for the patients because if they are already functioning well, there may be limited benefit and the potential for dissatisfaction following hip replacement. It might be supposed that mental health is affected by severe arthritis, and it would be improved after hip replacement. However, we found that the preoperative MCS score tended to be higher in participants undergoing hip replacement compared with controls with a small reduction in MCS score in participants following hip replacement, such that MCS scores were similar in participants with hip replacement postoperatively and the controls.

A strength of our study is that the study participants were drawn from a large, well- characterized, community-based clinical trial, which reduced the potential of bias in the selection of comparison group. With disability-free survival being the primary endpoint of the ASPREE trial, the participants were free of cardiovascular diseases and chronic disability at enrolment, reflecting the type of medically fit older individuals suitable for a hip replacement if clinically indicated. Thus, the ASPREE study offered us the ideal older population to examine the effect of hip replacement on functional outcomes. It was unbiased in terms of joint pain, but the ideal population to examine hip replacement. The approach of comparing participants with hip replacement with age- and sex-matched controls embedded in a large community-based clinical trial enables the functional outcomes of hip replacement to be investigated in the context of older adults with a similar likelihood of age-related functional decline. The health status profile in the current study sample was comparable to the wider ASPREE cohort where PCS scores declined with age and MCS scores tended to be stable with increasing age [36]. With the ASPREE trial being conducted across multiple states in Australia, participants underwent hip replacements in major public and private hospitals, with the decision to undergo hip replacement being made by the individuals and their healthcare providers as part of usual care. In the ASPREE study, all data were collected using standard, validated, well-described instruments used by trained observers following the published protocols to minimize measurement bias. Any potential differences across observers would have resulted in non-differential misclassification, so underestimate the study findings. We followed the Strengthening the Reporting of Observational Studies in Epidemiology (STROBE) Statement to report the study findings (STROBE checklist provided as a Appendix A). 

There are limitations in our study. As the time points at which baseline data on function were collected were different for participants with hip replacement and age- and sex-matched controls, the participants with hip replacement may have a different experience during the ASPREE trial, such as longer time on study medication and possibility of a higher risk of developing major health events (e.g., cardiovascular disease, dementia, cancer, or major bleeding), compared with the matched controls. This might affect the functional measurements and thus introduce bias. However, only participants who were medically fit would undergo a hip replacement whereas those who developed severe morbidities could not have a hip replacement. It is less likely that the accumulation of major health events would have affected the results of our study, as the results did not change in an additional analysis excluding the small proportion of participants who developed major health events prior to hip replacement (*n* = 62, 8.9% of the 698 participants with hip replacement). The time between baseline and follow-up outcome measures was longer in participants with hip replacement than in controls and this was adjusted for in the analyses. Of the original age- and sex-matched sample, 78% of the participants had completed health status data at follow-up. Although the follow-up rate was lower in participants with hip replacement (64%) compared with age- and sex-matched controls (93%), which might introduce selection bias, there were no significant differences in age, sex, BMI, morbidities or chronic conditions, and baseline measures of PCS, MCS, and gait speed between participants with follow-up data and those without. It is less likely that the loss to follow-up would have affected the results of the study. The ASPREE trial did not collect detailed joint-specific data as the study focused on the overall assessment of wellbeing. We were unable to determine the type of joint replacement which could have been primary or revision hip replacement. The results of our study did not change in sensitivity analysis excluding participants with self-reported joint replacement at baseline. It is also acknowledged that being volunteers in a long-term clinical trial, the ASPREE participants may have had a greater interest in their health.

## 5. Conclusions

In community-based older adults, although those receiving hip replacement experienced significant improvements in PCS score, their postoperative physical function remained significantly lower than age- and sex-matched controls. The greatest improvements were observed for bodily pain and physical function scores. Given that functional improvement is an important patient-related outcome after hip replacement, identifying those with poor preoperative physical function is most likely to identify older people most likely to benefit from hip replacement surgery. Further work is needed to optimize patient outcomes following hip replacement surgery. This will include identifying factors associated with better outcomes as well as ways to optimize selection of candidates for the surgery, including the timing of surgery and the risk–benefit ratio of hip replacement in community-dwelling older adults.

## Figures and Tables

**Figure 1 jcm-11-05117-f001:**
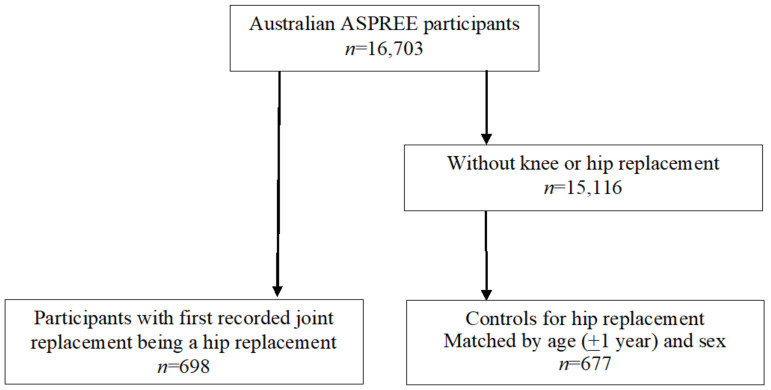
Participants with hip replacement and age- and sex-matched controls. ASPREE: ASPirin in Reducing Events in the Elderly.

**Table 1 jcm-11-05117-t001:** Characteristics of participants with hip replacement and age- and sex-matched controls at ASPREE baseline.

	Participants with Hip Replacement *n* = 698	Age-and Sex-Matched Controls *n* = 677	*p*
Age (matched), years	77.9 (4.3)	77.9 (4.3)	0.95
Females, *n* (%)	406 (58.2)	395 (58.4)	0.95
Education >12 years, *n* (%)	278 (39.8)	239 (35.3)	0.08
Body mass index, kg/m^2^	28.7 (4.7)	27.6 (4.7)	<0.0001
Diabetes mellitus, *n* (%)	66 (9.5)	68 (10.0)	0.71
Dyslipidemia, *n* (%)	486 (69.6)	452 (66.8)	0.25
Hypertension, *n* (%)	519 (74.4)	526 (77.7)	0.15
Chronic kidney disease, *n* (%)	145 (22.3)	187 (29.9)	0.002
Cancer, *n* (%)	117 (16.8)	132 (19.5)	0.40
Physical Component Summary score of SF-12	44.9 (9.8)	48.0 (8.9)	<0.0001
Mental Component Summary score of SF-12	56.3 (7.1)	55.7 (7.2)	0.14
Physical function	45.3 (10.2)	47.3 (10.1)	0.0003
Role physical	47.6 (8.9)	49.5 (8.3)	<0.0001
Bodily pain	45.7 (10.5)	50.1 (9.5)	<0.0001
General health	51.1 (7.5)	52.0 (7.4)	0.04
Vitality	53.7 (8.1)	54.7 (8.4)	0.03
Social functioning	53.2 (7.3)	53.6 (7.0)	0.22
Role emotional	51.4 (7.4)	51.5 (7.2)	0.79
Mental health	54.5 (7.6)	54.9 (8.1)	0.43
Gait speed, m/s	1.03 (0.23)	1.04 (0.23)	0.43

Data presented as mean (standard deviation) for continuous variables, and no (%) for categorical variables.

**Table 2 jcm-11-05117-t002:** Baseline and follow-up measures of health status and gait speed in participants with hip replacement and age- and sex-matched controls.

	Participants with Hip Replacement *n* = 445			Age- and Sex-Matched Controls *n* = 628			Difference between Groups	
	Baseline (Preoperative) Mean (SD)	Follow-Up Mean (SD)	Change Mean (95%CI)	Baseline (Study Entry) Mean (SD)	Follow-Up Mean (SD)	Change Mean (95%CI)	Change Mean (95%CI)	*p*
PCS	39.8 (11.0)	44.5 (10.8)	4.9 (4.0, 5.7)	48.2 (8.8)	48.0 (9.3)	0.01 (−0.7, 0.7)	4.8 (3.7, 6.0)	<0.001
MCS	56.9 (7.9)	55.6 (7.5)	−1.6 (−2.4, −0.8)	55.8 (7.1)	56.0 (6.6)	0.1 (−0.5, 0.8)	−1.7 (−2.8, −0.7)	0.001
Physical function	40.7 (11.4)	44.1 (11.6)	3.8 (2.9, 4.8)	47.6 (9.8)	47.7 (10.3)	0.2 (−0.6, 1.0)	3.6 (2.3, 4.9)	<0.001
Role physical	44.3 (9.8)	46.9 (9.6)	2.5 (1.6, 3.4)	49.7 (8.2)	49.7 (8.4)	0.2 (−0.6, 0.9)	2.4 (1.2, 3.5)	<0.001
Bodily pain	40.1 (11.2)	46.8 (10.8)	6.7 (5.6, 7.7)	50.2 (9.4)	49.9 (9.8)	−0.1 (−1.0, 0.8)	6.8 (5.4, 8.2)	<0.001
General health	49.4 (9.3)	50.3 (9.5)	0.7 (−0.1, 1.6)	52.2 (7.2)	52.0 (7.8)	−0.0004 (−0.7, 0.7)	0.7 (−0.3, 1.8)	0.17
Vitality	52.1 (9.5)	52.7 (9.2)	0.6 (−0.2, 1.5)	54.8 (8.3)	54.5 (8.1)	−0.2 (−0.9, 0.5)	0.8 (−0.3, 2.0)	0.14
Social functioning	51.5 (8.9)	52.4 (7.6)	0.7 (−0.1, 1.5)	53.7 (6.9)	54.0 (6.6)	0.2 (−0.5, 0.8)	0.5 (−0.6, 1.5)	0.36
Role emotional	50.4 (8.6)	50.3 (8.4)	−0.2 (−1.1, 0.7)	51.7 (6.9)	51.7 (7.5)	−0.03 (−0.7, 0.7)	−0.2 (−1.3, 1.0)	0.78
Mental health	54.2 (8.0)	54.3 (7.8)	0.1 (−0.8, 0.9)	55.0 (7.9)	55.4 (7.5)	0.5 (−0.2, 1.2)	−0.4 (−1.5, 0.7)	0.48
Gait speed (m/s) *	0.96 (0.25)	0.95 (0.22)	−0.02 (−0.04, 0.01)	1.05 (0.22)	1.01 (0.22)	−0.05 (−0.06, −0.03)	0.03 (0.0005, 0.06)	0.046

SD: standard deviation; CI: confidence interval; PCS: Physical component summary score of SF-12; MCS: Mental component summary score of SF-12. * *n* = 331 for participants with hip replacement and *n* = 574 for controls. Adjusted for baseline body mass index, education, morbidities or chronic conditions, and days between outcome measures.

**Table 3 jcm-11-05117-t003:** Baseline characteristics of participants with hip replacement based on minimal important change in self-reported physical function.

	Worsened PCS (PCS Change <−4.6) *n* = 69	Stable PCS (PCS Change −4.6 to 4.6) *n* = 168	Improved PCS (PCS Change ≥4.6) *n* = 208	*p*
Age at joint replacement, years	77.9 (4.7)	77.2 (3.9)	77.0 (4.3)	0.30
<75 years, *n* (%)	20 (29.0)	49 (29.2)	83 (39.9)	0.06
≥75 years, *n* (%)	49 (71.0)	119 (70.8)	125 (60.1)	
Females, *n* (%)	42 (60.9)	94 (56.0)	124 (59.6)	0.70
Education >12 years, *n* (%)	28 (40.6)	60 (35.7)	78 (37.5)	0.78
Baseline body mass index, kg/m^2^	28.1 (4.7)	28.6 (4.3)	28.1 (4.8)	0.51
Diabetes mellitus, *n* (%)	6 (8.7)	12 (7.1)	22 (10.6)	0.51
Dyslipidemia, *n* (%)	43 (62.3)	108 (64.3)	145 (69.7)	0.39
Hypertension, *n* (%)	51 (73.9)	128 (76.2)	148 (71.2)	0.54
Chronic kidney disease, *n* (%)	13 (20.6)	38 (25.2)	45 (23.0)	0.76
Cancer, *n* (%)	14 (20.3)	32 (19.1)	26 (12.6)	0.15
PCS	47.2 (8.9)	42.0 (10.9)	35.5 (9.8)	<0.0001
MCS	53.8 (8.0)	56.3 (7.2)	58.5 (8.1)	<0.0001
Physical function	46.9 (9.8)	42.6 (11.6)	37.0 (10.5)	<0.0001
Role physical	49.1 (8.2)	46.0 (9.2)	41.3 (9.9)	<0.0001
Bodily pain	46.7 (8.7)	41.9 (11.6)	36.5 (10.2)	<0.0001
General health	51.8 (7.4)	50.1 (9.4)	48.1 (9.7)	0.01
Vitality	52.6 (9.4)	53.6 (8.8)	50.7 (9.9)	0.01
Social functioning	52.6 (7.6)	51.3 (9.0)	51.3 (9.2)	0.53
Role emotional	50.1 (8.0)	50.0 (8.5)	50.9 (8.8)	0.60
Mental health	52.8 (8.1)	54.4 (7.9)	54.6 (8.0)	0.26
Gait speed, m/s	0.97 (0.22)	0.98 (0.25)	0.97 (0.26)	0.92
**Adjusted analysis ***				
PCS	47.6 (1.2)	42.5 (0.8)	35.8 (0.7)	<0.001
MCS	53.7 (1.0)	56.1 (0.6)	58.7 (0.6)	<0.001
Physical function	47.6 (1.3)	42.7 (0.8)	37.2 (0.7)	<0.001
Role physical	49.3 (1.1)	46.2 (0.7)	41.6 (0.7)	<0.001
Bodily pain	46.8 (1.3)	42.3 (0.8)	36.6 (0.7)	<0.001
General health	51.9 (1.1)	50.4 (0.7)	48.8 (0.6)	0.03
Vitality	52.3 (1.2)	53.6 (0.8)	50.9 (0.7)	0.03
Social functioning	52.8 (1.1)	51.6 (0.7)	51.5 (0.6)	0.58
Role emotional	50.3 (1.1)	49.7 (0.7)	51.0 (0.6)	0.38
Mental health	52.7 (1.0)	54.2 (0.6)	54.8 (0.6)	0.21
Gait speed, m/s	0.99 (0.03)	0.99 (0.02)	0.96 (0.02)	0.48

Data presented as mean (standard deviation) for continuous variables, and no (%) for categorical variables. PCS: Physical Component Summary score of SF-12; MCS: Mental Component Summary score of SF-12. * Adjusted for age, sex, body mass index, education, and morbidities or chronic conditions; data presented as mean (standard error).

## Data Availability

The data generated from this study will not be deposited in a public repository due to privacy and consent restrictions. De-identified, coded data can be made available from the corresponding author upon reasonable request, subject to a data sharing agreement.

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
