# Peer review of "Functional Outcomes Following Hip Replacement in Community-Dwelling Older Adults"

_jcm, 2022, doi:10.3390/jcm11175117_

Round 1

Reviewer 1 Report

1.      Keywords need to be reordered based on alphabetical order.

2.      What is the novel of the present study? Outcomes following hip replacement surgery have been widely studied in the literature. Nothing really new was brought into the existence of the present manuscript. The authors need to highlight their novelty in the introduction section.

3.      The sentence “Total hip replacement is one of the most successful surgical procedures, providing excellent pain relief and functional restoration for patients with severe hip OA” should adopt additional references published by MDPI to support this explanation as follows: Computational Contact Pressure Prediction of CoCrMo, SS 316L and Ti6Al4V Femoral Head against UHMWPE Acetabular Cup under Gait Cycle. J. Funct. Biomater. 2022, 13, 64. https://doi.org/10.3390/jfb13020064

4.      The discussion in the present manuscript is quite simple, expanding into more comprehensive discussion is mandatory.

5.      Further research needs to be explained in the conclusion section.

6.      Please use the Journal of Clinical Medicine, MDPI format properly. In the present form, the authors do not use it.

7.      The authors should proofread their manuscript due to grammatical errors and language style issues.

Reviewer 2 Report

First of all, the article must be tranfered to the template provided by the journal.

You need to present and organise your article using the Cohort checklist Microsoft Word - STROBE checklist cohort.doc (equator-network.org)

Please fill out the checklist and attach it to your paper as supplement.

Describe the validity and relibaility checkings process for the instrument. Have you assessed interrater relibaility for the data collection?

Please describe in detail the assignment of the participants to the groups.

Describe all measures taken for preventing bias in the process of research.

How about attrition of the participants in the groups and its impact on the results?

Round 2

Reviewer 1 Report

The authors has been fail to address the critical comments in the previous review report, especially numbers 2, 4, and 6. The revised form does not improve well and lacks quality. Nothing something really new with cutting-edge insight into community-dwelling older adults. The manuscript should be rejected and not published. The reviewer against this manuscript for publication. Thank you very much.

Reviewer 2 Report

Please incorporate your answers provided in your letter into the article final version. Your answers to my comments are accepted, but you need to make the changes in the text. Also cite STROBE checklist.
